# Process Management in IoT Operating Systems: Cross-Influence between Processing and Communication Tasks in End-Devices

**DOI:** 10.3390/s19040805

**Published:** 2019-02-16

**Authors:** Roberto Rodriguez-Zurrunero, Ramiro Utrilla, Alba Rozas, Alvaro Araujo

**Affiliations:** B105 Electronic Systems Lab, ETSI Telecomunicación, Universidad Politécnica de Madrid Avenida Complutense 30, 28040 Madrid, Spain; rutrilla@b105.upm.es (R.U.); albarc@b105.upm.es (A.R.); araujo@b105.upm.es (A.A.)

**Keywords:** operating system, IoT, process management, wireless communications, network stack

## Abstract

The emergence and spread of Internet of Things (IoT) technologies along with the edge computing paradigm has led to an increase in the computational load on sensor end-devices. These devices are now expected to provide high-level information instead of just raw sensor measurements. Therefore, the processing tasks must share the processor time with the communication tasks, and both of them may have strict timing constraints. In this work, we present an empirical study, from the edge computing perspective, of the process management carried out by an IoT Operating System (OS), showing the cross-influence between the processing and communication tasks in end-devices. We have conducted multiple tests in two real scenarios with a specific OS and a set of wireless protocols. In these tests, we have varied the processing and communication tasks timing parameters, as well as their assigned priority levels. The results obtained from these tests demonstrate that there is a close relationship between the characteristics of the processing tasks and the communication performance, especially when the processing computational load is high. In addition, these results also show that the computational load is not the only factor responsible for the communication performance degradation, as the relationship between the processing tasks and the communication protocols timing parameters also plays a role. These conclusions should be taken into account for future OSs and protocol developments.

## 1. Introduction

The current projections of IoT are largely due to the last two decades of research in Wireless Sensor Networks (WSNs). However, the unique characteristics of IoT devices and applications, and its network architecture, represent a paradigm shift that poses many new and important challenges to overcome.

One of these challenges is to prevent the growth in the number of devices and connections in the coming years from becoming an increase of similar proportions in the Internet Protocol (IP) traffic and spectrum occupancy. On the one hand, this fact has direct implications for communication management. In a scenario marked by the increasing spectrum scarcity problem, communications must reach a high level of efficiency and dependability. They should avoid interferences or bottlenecks in the network that can increase packet loss and, therefore, cause retransmissions and affect network traffic.

On the other hand, IoT applications increasingly demand a more comprehensive understanding of the environment. That is, applications are no longer limited to obtaining direct measures of certain environmental variables through simple sensors, e.g., temperature, humidity or light sensors, but a higher level of information is sought through the processing of these raw data [1,2,3]. Performing this processing in the cloud is not desired because of latency issues and the generation of a massive network traffic due to the data uploading. For this reason, and because of the improvement of features and resources in the end-devices, these devices are increasingly becoming responsible for all or some of this processing. This trend of placing the computing and storage capabilities where the data is generated is referred to as edge and/or fog computing and it is gaining importance in the IoT [4,5]. However, delegating these storing and processing tasks to the end-devices increases again their computational load, and introduces other challenges in terms of resource and process management: real-time scheduling, storage allocation, energy consumption, among others.

This leap of complexity with respect to the WSNs has made the use of an OS something practically assumed, even in end-devices. The OS is the one that ultimately carries out the management of all the above aspects: communications, resources and processes. Moreover, this management is performed according to certain OS configurations and policies, and it also depends on multiple aspects of implementation, e.g., data structures, threading and programming models, buffer sizing and handling. All these characteristics influence the final performance of devices and are correlated with each other to a greater or lesser extent. However, as detailed in the next section, many reference works, especially in the field of WSNs, ignore these cross-effects based on assumptions that are no longer so obvious in the IoT field. This work is motivated by this fact and by our observation of these cross-effects in the development of real IoT applications in research projects. These cross-effects imply that, processing tasks can have a large impact on the communication performance, while, on the other hand, the execution of communication tasks may imply that the time constraints of the processing tasks are not met. 

Consequently, this work aims to approach this interdisciplinary research area through an empirical study. Specifically, on the one hand, our contribution is a quantitative evaluation of the influence that the OS management of running processing tasks has on the performance of wireless communications. This has been achieved by conducting multiple tests with different levels of computational load, various priority schemes, single-hop and multi-hop scenarios, etc. On the other hand, we have also evaluated the influence that a communication parameter—the radio duty cycle—has on the use of system resources and, therefore, on the performance of other tasks that may require them. 

This paper is organized as follows. Section 2 presents related work in OSs for IoT as well as studies that analyze the performance of low-power wireless communications. Section 3 contains a full description of the hardware and software resources employed in this empirical study, as well as the methodology used. Section 4 presents the results obtained in the different tests, which are discussed in Section 5. Finally, conclusions are offered in Section 6.

## 2. Related Work

OSs for wireless sensor devices have been developing for years as they are considered a key element for the IoT [6]. The first to appear were TinyOS [7] and, subsequently, Contiki OS [8]. Both were developed for devices with very limited resources, so they have an event-driven scheduler that limits the development of applications with real-time restrictions. Later on, these OSs have incorporated additional libraries that provide the ability to execute preemptive threads, although with some limitations, as they do not have task priorities or task synchronization tools. 

In recent years, the increase of available resources in wireless devices has made it possible to use OSs such as RIOT [9], Mbed OS [10] or FreeRTOS [11], which include preemptive priority-based task schedulers and provide full real-time support. They also provide various features to facilitate IoT application development and improve resource management. Therefore, although the most recent OSs are different from each other in terms of capabilities and features, they share a similar process management as their core element.

On the other hand, many of these OSs provide stacks of protocols responsible for managing the wireless communications of the devices. Since the appearance of the first WSNs, researchers have focused on the study of network stacks, which has led to the development of a large number of wireless protocols [12] and even some standards (IEEE 802.15.4, 6LoWPAN, ISA-100.11a). As a result, there are a variety of protocols for the MAC [13,14,15], routing [16,17], and transport [18] layers developed to overcome the different issues present in WSN communications, such as energy-efficiency, latency, scalability, throughput or quality of service (QoS). Moreover, several modifications have been made to IP-based protocols to better fit WSN devices with limited resources, and other application protocols such as CoAP have also been designed to integrate those devices into the public Internet. 

In addition, numerous simulations and empirical studies have been carried out to evaluate the performance of the different protocol layers and end-to-end wireless communications. It is worth highlighting the work of K. Srinivasan et al. [19] in which they provide a comprehensive empirical study that characterizes low-power wireless communications at the link level. In this study, the authors present different Packet Reception Rates (PRRs) for different scenarios, as well as the causes that produce the reception rate to be time variable and the links to behave asymmetrically, providing a practical basis for the development of new protocols. On the other hand, several studies compare the performance of different MAC protocols [20], and others analyze the throughput, latency or reliability of specific protocols, such as CSMA-MAC [21] or the IEEE 802.15.4 standard [22]. Other works have focused on characterizing IP-based communications for WSN nodes. For example, the study by Sumeet Thombre et al. [23] provides measures of latency, packet error rate and throughput using a complete network stack and the Contiki OS. 

Nevertheless, most of these simulation works and empirical studies ignore the software implementation of communication protocols and the effect that other tasks running simultaneously in the operating system may have on the communication performance. Traditionally, it has been assumed that the main task of the nodes is to sense the environment and directly transmit that data to a sink node. However, as stated before, edge and fog computing approaches are gaining importance [4,5], and nodes are expected to carry out more processing tasks than traditional sensor nodes. Therefore, as far as we know, the existing literature lacks detailed analyses of the influence that those processing tasks have on the performance of wireless communications. Thus, the increase in the computational load of the nodes makes it necessary to evaluate this influence and the cross-effects that exist between the processing and communication tasks.

## 3. Materials and Methods

This work aims to carry out a study of the influence of the processing tasks in the performance of wireless communications, and vice versa. As seen in the previous section, there is a wide variety of OSs and protocols for WSNs in the literature. This makes it very difficult to evaluate the cross-effects in all of them that would allow us to draw generalizable conclusions for all OSs and protocols. Therefore, as a first approach, we propose an empirical study with a specific OS and a specific stack of protocols. However, the results obtained may be applied to other OSs or protocols if their characteristics are more or less similar. In this way, it can be used as a basis for future broader studies to generalize the studied cross-effects and facilitate the development of future wireless communication protocols and OSs features for IoT systems.

### 3.1. Hardware

As physical support for all the tests carried out in this work, the custom-developed YetiMote IoT devices presented in Figure 1 [24] have been used. These nodes are based on a high-performance low-power STM32L476RE microcontroller [25] running at 48 MHz. This microcontroller is internally equipped with 512 KB of Flash memory and 128 KB of RAM memory, and supports several low-power modes. Several sensors (accelerometers and temperature) are provided in YetiMote nodes as well as several Input/Output interfaces: UART, SPI, I2C, USB and a microSD card slot. Therefore, these nodes have many more resources than the nodes traditionally used in WSN—such as Micaz or TelosB. This makes them suitable to run computing-intensive processing tasks as expected in some IoT applications in which sensor devices must provide processed high-level information. For this reason, the experimental tests carried out in this work are performed using YetiMote nodes that run processing and communication tasks.

These nodes also include an 868 MHz radio interface, which consists of a SPIRIT1 low-power transceiver [26], used with the following configuration:Modulation scheme: GFSK.Receiver bandwidth: 540 kHz.Frequency deviation: 127 kHz.Bit rate: 250 kbps.Output power: 11 dBm.Automatic CRC handling: 1 byte.Preamble: 4 bytes.Synchronization Word: 4 bytes.Forward Error Correction (FEC) enabled.Data whitening enabled.

### 3.2. Software

The tests performed in this empirical study have been done using YetiOS [27]. This OS is built on top of FreeRTOS, providing several extra features such as a linux-like device drivers, timing and standard Input/Output interfaces, and an adaptive core which allows the OS to improve its performance in dynamic environments. In addition, YetiOS maintains the same preemptive round-robin priority scheduler as FreeRTOS. The increase of available resources in low-power wireless devices—such as YetiMote nodes—allows us to implement this FreeRTOS-based OS, as well as other recent OSs used in IoT sensor devices such as RIOT or Mbed OS. This way, we have used YetiOS since it provides all the features needed to carry out the tests of this work, and its process management—the same as FreeRTOS—is comparable to that of other OSs, such as RIOT and Mbed OS. 

YetiOS also provides a layered network stack. Each layer is implemented independently, providing a useful framework for protocol designers. For this work, we have used custom-developed MAC, routing and transport layers. The MAC layer (YetiMAC) uses a simple Time Division Multiple Access (TDMA) scheme to access the wireless channel, as presented in Figure 2. In YetiMAC, the time is divided into frames of $T_{MAC}$ duration, which are themselves divided into two parts: the time allocated to send and receive packets $T_{packets}$, and the time allocated to send and receive beacons $T_{beacons}$. This way, in general, the node is listening for incoming beacons during $T_{beacons}$ time while it is in sleep state the rest of the time. During $T_{packets}$, the node listens to packets only if it has received a beacon to wake up on the current frame. Therefore, when the node is not receiving or transmitting packets, the Radio Duty Cycle (RDC) is the quotient of the beacon time by the frame time as presented in (1).
(1)$$\mathrm{RDC}=\;\frac{T_{beacons}}{T_{MAC}}$$

In addition, that MAC allows dividing both the beacon and packet time in different slots in order to reduce collisions when several nodes transmit during the same frame. The larger the number of slots, the lower the probability of packet collisions. As shown in Figure 2, when a packet is scheduled to be transmitted, a beacon is sent in a free slot of the beacon time. Then, the packet is randomly allocated within the corresponding slot (the same slot number as the one of the beacons that was sent) and is transmitted to the receiver node, which is in a listening state after having received the previous beacon. It is also important to note that the nodes are required to be synchronized for this MAC scheme to work, so broadcast synchronization packets are periodically sent. In this MAC, all nodes have the same role, except for the one that acts as the gateway, which is always in a listening state and is the source of the synchronization packets. 

Although we use a custom-defined MAC to perform our tests, this MAC is similar to others typically used in wireless networks. Most of them follow scheduled schemes with strict timing constraints—such as the standard 802.15.4e that uses Time-Slotted Channel Hopping (TSCH) scheme—which makes the process management a critical issue. It is important to note that on these scheduled MAC schemes, the communication performance may drastically degrade if these timing constraints are not met.

As for the routing layer, multi-hop mesh routing is implemented. Sensor nodes periodically send flooding packets in order to self-discover the best route to reach each node in the network. All the nodes have the same role and they may forward packets when they are not the final destination.

With regard to the transport layer implemented, it provides both a connectionless unreliable protocol—similar to User Datagram Protocol (UDP)—and a connection-oriented reliable communication protocol—similar to Transmission Control Protocol (TCP)—, although we only use the first one in our tests. This protocol provides the functions that may be used in the application layer to send and receive packets on a specific port.

The network stack described requires several tasks to perform the operations of all the layers. All these tasks except the one in charge of synchronization have the same priority in our system and will be changed together for the different tests. The one in charge of processing the synchronization packets is fixed and always has the highest available priority. In addition, a reference task is also included in our test system to emulate the behavior of a processing task in sensor devices. We have defined this reference task as presented in Figure 3, a periodic task with period $T_{PROC}$. It is executed in $T_{ON}$ time and spends the rest of the time, $T_{OFF}$, in an idle state. Therefore, we can define the Processing Task Duty Cycle (PTDC) as presented in (2).
(2)$$\mathrm{PTDC}=\;\frac{T_{ON}}{T_{PROC}}$$

### 3.3. Methodology

Once we have detailed the hardware and software resources we have used in this empirical study, we explain the methodology followed in our tests in this subsection. We have used two basic scenarios as presented in Figure 4. In the single-hop scenario, we only use two nodes, node N1 being the one in charge of periodically transmitting packets to node GW, which works as the gateway (that means the MAC is always in a listening state). In the multi-hop scenario, node N2 has enough radio coverage to reach nodes N1 and GW, but these two nodes do not have radio coverage between them. Therefore, in this case, the packets that N1 periodically sends to GW are routed through N2. We have chosen simple scenarios to focus on the influence that the processing tasks have on the communication performance, isolating it from other effects that may occur in deployments with more nodes, such as collisions, multipath effects, routing or spectral saturation.

Next, we are going to present the experimental conditions of both scenarios. Since there are many configurable parameters in our system, we set some of them to fixed values while we vary the others in the different tests to determine their influence.

With regard to the MAC layer, we set the number of packet slots to 3 for all tests since it is an acceptable value for a scenario in which many collisions are not expected. As a result, the value of $T_{beacons}$ is always 13 ms, as it is the time used by the 3 beacon slots plus the guard time required to face tiny desynchronizations. In addition, the RDC value may be 10%, 15% or 20% depending on the running test, which means that the $T_{MAC}$ and $T_{packets}$ times change for each value of RDC as presented in Table 1. These values have been chosen since they are representative of typical radio duty cycles in energy-constrained sensor networks. In each test, all the nodes in the scenario have the same RDC value.

As for the processing tasks, we define 3 different types—TT1, TT2 and TT3—with an execution time $T_{ON}$ of 2 ms, 5 ms and 50 ms respectively. In addition, we perform the tests varying the PTDC, which can take the value of 5%, 25% or 50%. Therefore, the $T_{OFF}$ and $T_{PROC}$ times change for each PTDC and task type according to Equation (2). Table 2 summarizes all the possible configurations of the processing task. It is very important to highlight that this processing task is only running in node N1. Nodes N2 and GW do not have this reference task running, they only route and receive packets respectively.

In our tests, we also change the priorities of the communication tasks of the network stack and the processing one. However, as said before, the synchronization task has a fixed priority, which is the highest possible. This task is executed when a synchronization packet is received (once every 10 s), and runs for less than 100 µs. Besides, there are other auxiliary tasks running in the OS (stdio task, tracing task), but their priority is the lowest possible and their combined CPU load is much lower than 1%. Therefore, we assume that the influence of both the synchronization and the auxiliary tasks can be considered negligible in our tests. This way, we define 3 different priority schemes, PS1, PS2 and PS3, by setting different priority levels to the communication tasks and the reference processing task. In PS1, the communication tasks have higher priority level than the processing task, while in PS2 they have the same priority level. Finally, in PS3, the processing task have higher priority level than communication tasks. Table 3 presents the priority levels for the different priority schemes, 1 being the lowest priority and 4 the highest possible one. As stated before, the processing task only runs in node N1, so these priority levels are always constant in nodes N2 and GW. For these nodes, the communication tasks have a priority level of 3 for all priority schemes.

Each test performed for each scenario lasts 3 min. In each one of them, node N1 is continuously sending packets from the top of the network stack, so all the layers described before are involved in the communication. In this way, each packet takes a variable time to be transmitted, which depends on the MAC schedule and the transceiver configuration. Then, when a packet has been sent, the node waits for 100 ms to start sending the next packet, and so on until the test time is over. Therefore, the time between packets is defined by the sum of the variable transmission time and this fixed wait time. It is considered that node GW has received a packet when it reaches the top of the network stack. Node N2 does not send or receive any packet from its application layer, it just forwards packets from its routing layer.

Each packet sent is filled with 40 application bytes and contains a unique and incremental identifier. This is the maximum application payload length allowed by the network stack, since up to 24 bytes are reserved for protocol headers and the maximum buffer size of many radio transceivers is 64 bytes. The incremental identifier allows us to count the number of received and missed packets in the GW node. In addition, a timestamp is also included, so we can also obtain the latency of each packet when it is received, since all the nodes are synchronized. In the GW node, we also calculate the throughput at the end of each test by dividing the number of application bytes received by the test time. We consider this latency and throughput as our communication performance metrics.

Finally, we also monitor the running tasks in node N1 by using Percepio Tracealyzer software [28]. This tool allows us to obtain the execution parameters of the tasks running in the sensor node, such as the execution time, the wait time and the turnaround time. Specifically, the wait time is the metric we have used to evaluate the effect that the communication parameters have in the processing task, since it is the time the task is blocked by other tasks running in the OS.

Summarizing, we have performed a set of experimental tests by changing the RDC, the PTDC, the task type and the scheduling priorities. For each test, we obtain the throughput, the latency and the wait time of the reference processing task. This way, we evaluate the influence that the PTDC and the processing task type have in the communication performance. In addition, we also evaluate the process management influence by changing the priority scheme. In the opposite direction, the RDC communication parameter is changed to evaluate its influence on the processing task wait time.

## 4. Experimental Results

In this section, we present the results obtained from the tests performed in this work. These results are also published in a public dataset [29], so that additional analyses can be carried out if necessary. Figure 5, Figure 6, Figure 7 and Figure 8 show the most relevant ones. In all of the figures, the bars in blue tones correspond to the data obtained using the priority scheme PS1, while the ones in orange and green tones represent those of the priority schemes PS2 and PS3, respectively. In addition, within the same color, the different tones correspond to the different types of the processing task used in the tests. Specifically, the lightest ones represent the tests performed with TT1, the medium tones with TT2 and the dark ones with TT3. All this information is summarized in the legend of each figure to facilitate its understanding.

### 4.1. Single-Hop Scenario

As mentioned before, this scenario consists of a single-hop link between a sensor node N1 and the gateway node GW. It should be remembered that N1 is also running a parallel processing task.

The data in Figure 5 correspond to the measurements obtained with the RDC of the devices set at 10%, which is a typical reference value in WSNs. Specifically, the two upper graphs represent the average latency and throughput of the link for the different PTDC values. Moreover, the average wait time of this task is also shown in the last graph of the figure. 

The most noteworthy feature of the two upper graphs is the fact that with priority schemes PS1 and PS2 the communication performance remains stable even if the PTDC increases. This occurs for the three task types evaluated. However, in the PS3 scheme, the communication tasks have a lower priority than the processing one. In this case, the latency increases and the throughput decreases as the PTDC gets higher. Moreover, the longer the $T_{ON}$ time of the processing task, the greater the magnitude of this effect. Finally, in the last graph, we can observe an opposite effect in the wait time of the processing task. With PS3—communications having a lower priority than the processing task— these times are very low and stable regardless of the PTDC but, with the other priority schemes, the wait time increases very significantly. In this case, the task type, i.e., the different execution times $T_{ON}$, also causes large differences in the results.

In Figure 6, the latency, the throughput and the average processing task wait time are again represented, but in this case the PTDC is set at 5% and is the RDC of the devices what varies.

In this way, we can see the effect of the RDC on the communication performance, which is quite intuitive to understand. However, it is worth noting the effect that this communication parameter has on the wait time of the processing task, even when it has a very low duty cycle (5%). This can be seen in the lower graph, where we can observe that, for PS1 and PS2 (when the priority of the processing task is lower or equal to that of the communications), the wait time of the processing task increases with the RDC. In this case, again, the type of task causes great differences in the results.

### 4.2. Multi-Hop Scenario

This scenario consists of a link between a sensor node N1 and a gateway node GW through another sensor device N2. In this case, N1 is also running a parallel processing task, but N2 only acts as a router between the other two. Figure 7 and Figure 8 are analogous to Figure 5 and Figure 6 respectively, but with the results of the multi-hop scenario.

In Figure 7, we can observe how, as happened in the single-hop scenario, the communication performance decreases with the PTDC when PS3 is implemented, while it remains stable with PS1 and PS2. However, it should be noted that in this case, this effect does not have a direct relationship with the processing task type, i.e., its execution time. While in the single-hop scenario, the lowest value of the throughput corresponded to TT3, in this case it occurs with TT2, practically reaching a communication loss when the processing load is high (50%). As it will be detailed in the discussion, this reveals that what is critical is not the $T_{ON}$ time itself (that only depends on the task type), but the relationship between the processing tasks times and the TDMA scheme of the MAC layer. Finally, in the lower graph of the figure, the observed behavior of the wait time is analogous to that of the single-hop scenario.

In Figure 8, as was the case in the single-hop scenario, we can see that for PS1 and PS2, the wait time of the processing task increases with the RDC and that the type of task causes great differences in the results. In addition, regarding the two upper graphs, we observe that the RDC value has a greater impact on the communication performance than it had on the single-hop scenario.

## 5. Discussion

As the process management carried out by the OS is responsible for both the processing and the communication tasks, these are not independent of each other. In this way, we have confirmed empirically that there are multiple factors related to both types of tasks that generate cross-effects between them to a greater or lesser extent. 

The main conclusions extracted from the results presented in the previous section are:When the priority of the communication tasks is greater than that of the processing task, the performance of the communications is not significantly affected, regardless of the level of computational load associated with said processing task.When the priorities of both types of tasks are equal, the scheduler allocates processor times with a simple round-robin policy which does not take priority into account. Despite what might be expected, this still has no significant effect on the performance of communications, even for high processing task duty cycles.However, if the processing task has a lower or equal priority than the communication tasks, its wait time increases significantly, especially if its execution time ($T_{ON}$) is high. This can be critical for tasks with strict timing constraints. Moreover, it has been shown that the RDC also affects the processing task wait time. Specifically, if the RDC is increased to improve the communication performance, the wait time of the processing task correspondingly increases.

On the other hand, when the communication tasks have a lower priority than the processing one, there is an important impact in the communication performance in two ways.
First, when the processing task duty cycle increases the communication performance is highly degraded, which means that the throughput is reduced and the latency is increased, especially in the multi-hop scenario.Finally, we have observed that this degradation of the communication performance is highly dependent on the processing task type, i.e., on its execution $T_{ON}$ time. This degradation of the communication performance is due to the relationship between the TDMA scheme imposed by the RDC, and the processing task times, imposed by its type and its PTDC. In this way, under certain conditions, the communication tasks are not able to transmit the packet in its allocated time slot. Therefore, the TDMA scheme is displaced and the receiving node is not in a listening state when the packet is transmitted. As seen in the results, this effect can reach the point of a total loss of communication.

In summary, we have proven that there are multiple trade-offs that must be taken into account for the effective implementation of edge computing, especially from the perspective of low-resource end-devices. Traditional communication schemes, based mostly on TDMA or hybrid strategies, have strict time requirements that, if not met, greatly degrade the performance of communications. However, this is in contrast with the idea of edge computing, since the implementation of certain tasks with real-time restrictions and high CPU usage can have harmful effects on these types of communication schemes.

At this point, it is very important to highlight that in some situations the cross-effects presented may result in a complete misfunction of an IoT application. For example, a healthcare or surveillance application may require an intensive processing task with strict timing constraints that must have a high priority. Under these conditions, the communication performance can degrade drastically to the point of breaking the communication link, thus preventing the sending of alerts. Otherwise, if the communication is prioritized, it may result in large wait times on the processing task, so it could not meet its timing constraints. In addition, our results have proven another important problem for these applications. The performance degradation in some situations is highly variable depending on what we call the task type, i.e., the task executing time $T_{ON}$. This implies that the cross-effects are unpredictable depending on the tasks needed for each application, which complicates setting homogeneous guidelines for application developers that use the OS.

Therefore, these challenges must be faced at all levels of development. At the hardware level, an increase in computational resources in low-power devices could reduce the negative effects in communications when the CPU load is high. In fact, there are already low-power microcontrollers that include a second CPU, which may be exclusively dedicated to communications [30]. On the other hand, at the software level, it would be necessary to design communication protocols that take into account this high processing load. In addition, new process management methods or software strategies could also be developed that allow the OS to jointly manage the communication protocols and the scheduling of the processing tasks in order to comply with the restrictions of both. Finally, hybrid strategies could also be designed to make the boundary between hardware and software more flexible and, in this way, implement the tasks with stricter time constraints at a hardware level, thus allowing the most critical actions to be parallelized.

## 6. Conclusions

The evolution of WSNs towards the IoT in general, and the edge and fog computing approaches in particular, has had a significant impact on the computational load of the end-devices. As a result, current and future sensor devices are expected to provide more highly-processed information compared to traditional WSN nodes, which highly increases their computational loads.

In this work, we have carried out an empirical study of the cross-influence that the computational load of processing tasks has in the communication performance when using an OS for IoT sensor devices. The main conclusion drawn from this work is that the increase in computational load results in the appearance of a series of cross-effects between the processing and communication tasks that can significantly affect their performance. It is important to emphasize that this circumstance can cause the complete malfunctioning of an application. Future designs must therefore take into account both aspects in order to reach a compromise between their requirements, especially when both the processing and communication tasks have strict timing constraints. This poses a series of new challenges that will have to be faced at multiple levels: OS process management, communication protocols, hardware architecture of end-devices, microcontroller features, etc.

To draw these conclusions, in this empirical study real IoT sensor devices have been used to conduct multiple tests with different levels of computational load, various priority schemes, different scenarios, etc. This study was performed with a specific set of tools: an OS with a round-robin priority scheduler and a layered network stack with a TDMA-based MAC. These features are present in many other IoT sensor devices, so the results can be extrapolated to other cases in which the characteristics of the OS and the communication protocols are similar.

However, it is convenient to go deeper into this study to better understand the influence on the observed effects of other aspects that, due to the scope of this preliminary work, have not been considered. Along these lines, it would be interesting to experiment with a connection-oriented reliable communication protocol (TCP-like), different MAC schemes, multiple payload lengths, more complex scenarios with a greater spectrum occupancy, etc. In addition, it would also be useful to replicate these experiments with other reference IoT OSs and platforms. The realization of these studies will constitute a necessary basis for the development of new mechanisms to reduce the cross-effects between the processing and communication tasks.

## Figures and Tables

**Figure 1 sensors-19-00805-f001:**
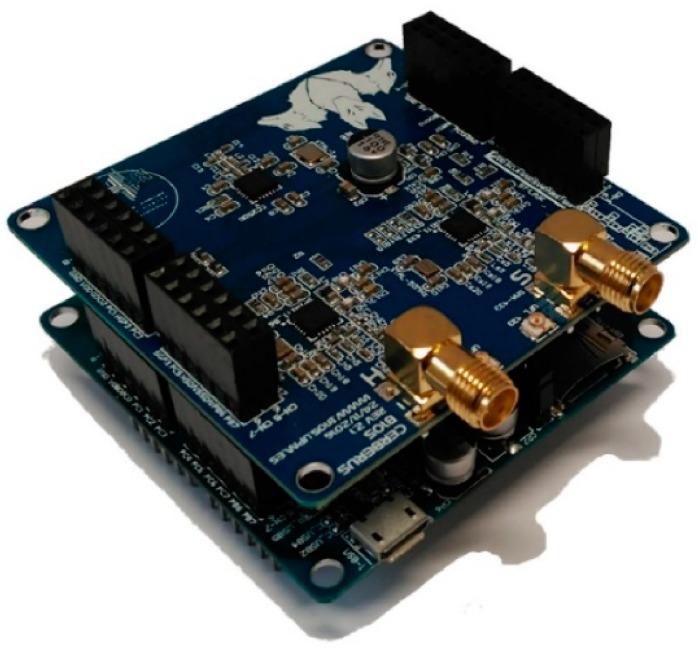
YetiMote Node.

**Figure 2 sensors-19-00805-f002:**
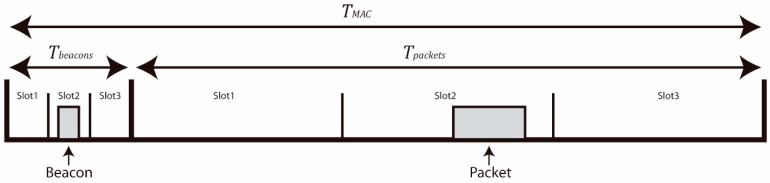
MAC frame timing scheme.

**Figure 3 sensors-19-00805-f003:**
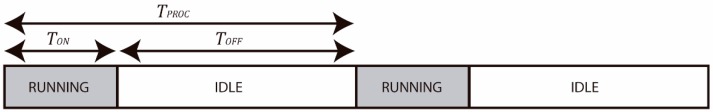
Execution scheme of the reference task.

**Figure 4 sensors-19-00805-f004:**
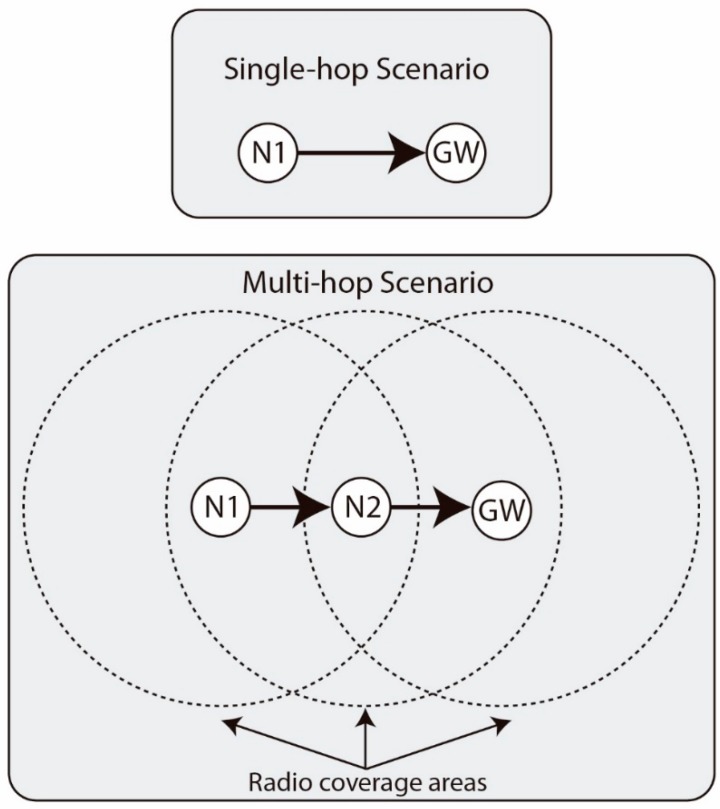
Single-hop and multi-hop scenarios implemented in our tests.

**Figure 5 sensors-19-00805-f005:**
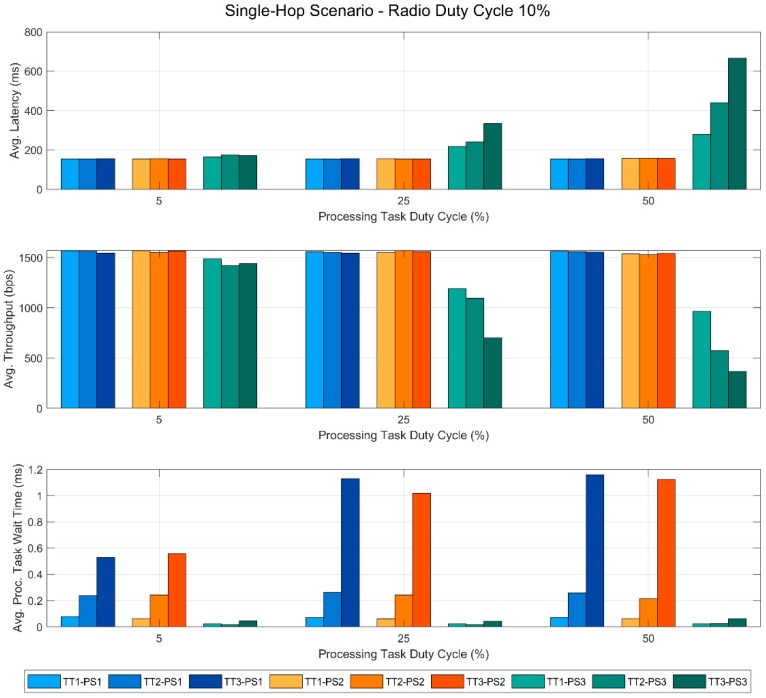
Single-hop scenario with a fixed RDC of 10%.

**Figure 6 sensors-19-00805-f006:**
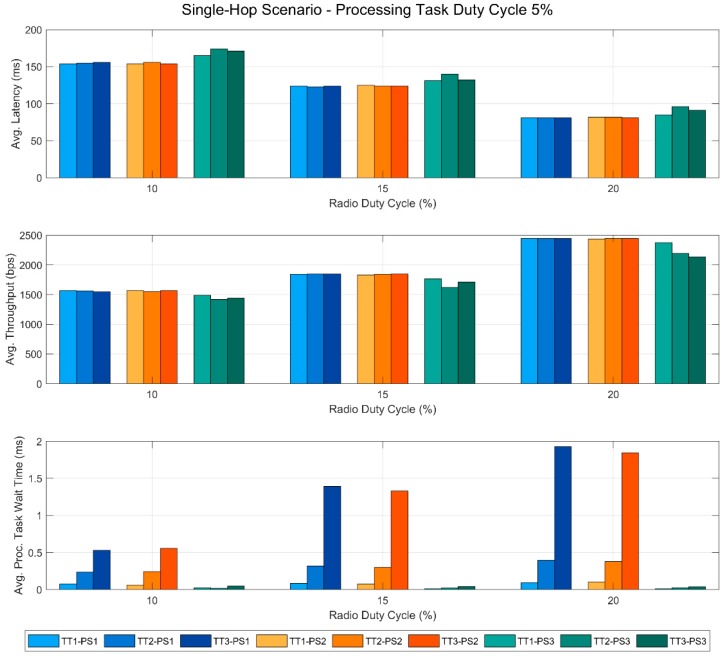
Single-hop scenario with a fixed Processing Task Duty Cycle of 5%.

**Figure 7 sensors-19-00805-f007:**
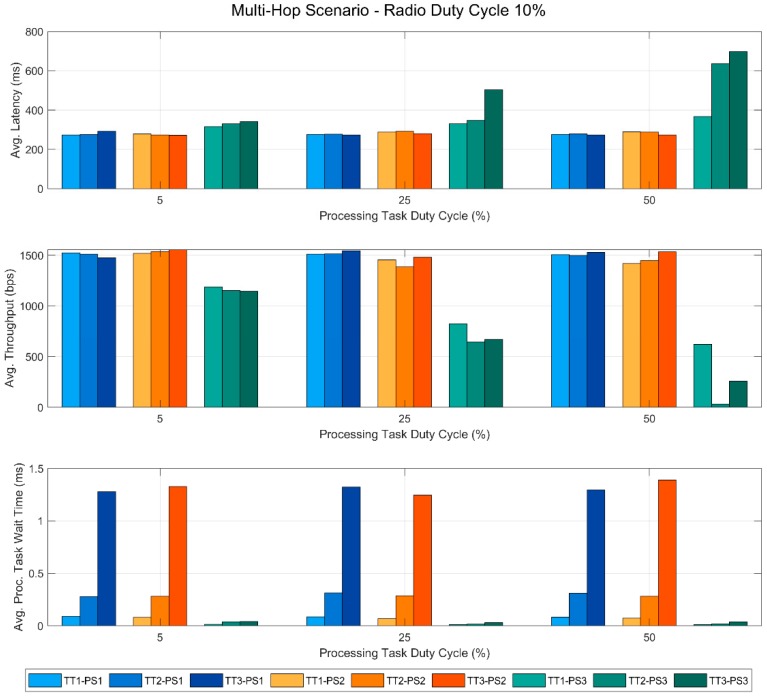
Multi-hop scenario with a fixed RDC of 10%.

**Figure 8 sensors-19-00805-f008:**
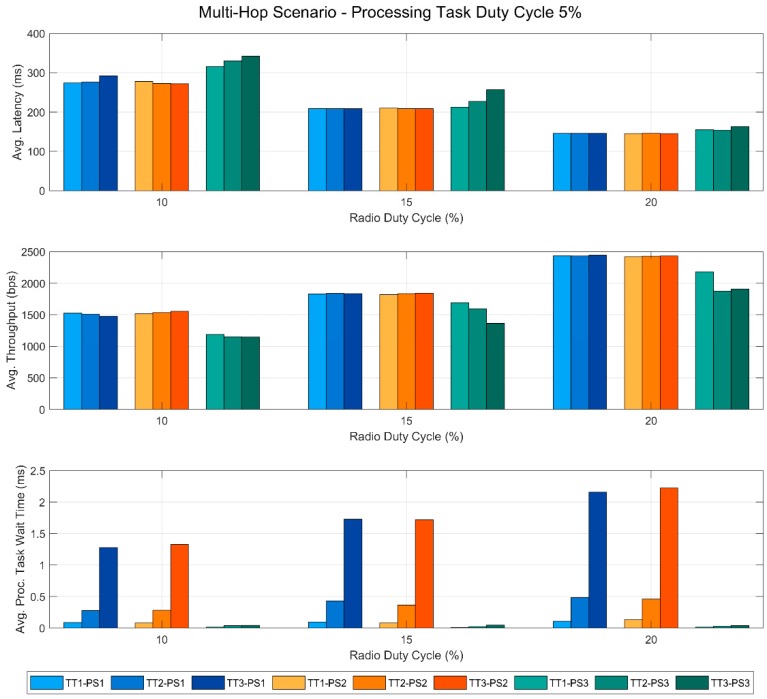
Multi-hop scenario with a fixed Processing Task Duty Cycle of 5%.

**Table 1 sensors-19-00805-t001:** MAC times for different RDC values.

	$T_{beacons}=13\;ms$	
RDC	*T_MAC_*	*T_packets_*
10%	130 ms	117 ms
15%	87 ms	74 ms
20%	65 ms	52 ms

**Table 2 sensors-19-00805-t002:** Processing task times for different PTDC and task type values.

PTDC	Task Type	*T_ON_*	*T_PROC_*	*T_OFF_*
5%	TT1	2 ms	40 ms	38 ms
	TT2	10 ms	200 ms	190 ms
	TT3	50 ms	1000 ms	950 ms
25%	TT1	2 ms	8 ms	6 ms
	TT2	10 ms	40 ms	30 ms
	TT3	50 ms	200 ms	150 ms
50%	TT1	2 ms	4 ms	2 ms
	TT2	10 ms	20 ms	10 ms
	TT3	50 ms	100 ms	50 ms

**Table 3 sensors-19-00805-t003:** Priority levels of N1 node tasks for the different priority schemes.

	Priority Levels (1–4)		
Tasks	PS1	PS2	PS3
Communication tasks	3	3	2
Processing task	2	3	3
Synchronization task	4	4	4
Auxiliary tasks	1	1	1

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
