# Peer review of "Process Management in IoT Operating Systems: Cross-Influence between Processing and Communication Tasks in End-Devices"

_sensors, 2019, doi:10.3390/s19040805_

Reviewer 1 Report

The paper is very well-written, and concepts are adequately motivated and clarified. Conclusions are supported by the obtained results and are significant in the emerging field of edge computing. Some minor concerns must be improved:

·       Authors must motivate the choice of the YetiMote and software as device for the conducted tests.

·       Authors should point out possible future works in the conclusions.

Author Response

The authors would like to thank the reviewers for their time and their valuable comments. We really appreciate their effort and interest.

Following the reviewer 1’s suggestions and comments, we have made the following changes to our paper.

The paper is very well-written, and concepts are adequately motivated and clarified. Conclusions are supported by the obtained results and are significant in the emerging field of edge computing. Some minor concerns must be improved:

Point 1: Authors must motivate the choice of the YetiMote and software as device for the conducted tests.

Response 1: YetiMote nodes implements a last-generation ARM Cortex-M4 low-power microcontroller—STM32L476—. This microcontroller provides a larger computational capability than others traditionally used in energy-constrained devices, such as the ones present in Micaz and TelosB platforms. For this reason, YetiMote nodes are used to perform the tests of our work, since they allow implementing computing-intensive and delay-sensitive IoT applications. As for the software, YetiOS—which is based on FreeRTOS—provides advanced features while fitting in last-generation platforms, and its process management is comparable to that of other typically used IoT OSs. It is expected that the conclusions drawn in this work are similar if other platform is used, since there is no aspect in the choice of the microcontroller and the software that is decisive.

In order to clarify that point, we have included the following text in Section 3:

“Therefore, these nodes have many more resources than the nodes traditionally used in WSN—such as Micaz or TelosB—. This makes them suitable to run computing-intensive processing tasks as expected in some IoT applications in which sensor devices must provide processed high-level information. For this reason, the experimental tests carried out in this work are performed using YetiMote nodes that run processing and communication tasks. “

“The increase of available resources in low-power wireless devices—such as YetiMote nodes—allows us to implement this FreeRTOS-based OS, as well as other recent OSs used in IoT sensor devices such as RIOT or Mbed OS. This way, we have used YetiOS since it provides all the features needed to carry out the tests of this work, and its process management—the same as FreeRTOS—is comparable to that of other OSs, such as RIOT and Mbed OS. “

Point 2: Authors should point out possible future works in the conclusions.

Response 2: We have added in the Section 5 – Conclusions the following paragraph reflecting on future directions of the work:

“However, it is convenient to go deeper into this study to better understand the influence on the observed effects of other aspects that, due to the scope of this preliminary work, have not been considered. Along these lines, it would be interesting to experiment with a connection-oriented reliable communication protocol (TCP-like), different MAC schemes, multiple payload lengths, more complex scenarios with a greater spectrum occupancy, etc. In addition, it would also be useful to replicate these experiments with other reference IoT OSs and platforms. The realization of these studies will constitute a necessary basis for the development of new mechanisms to reduce the cross-effects between the processing and communication tasks.”

Thank you for your collaboration and do not hesitate to contact us if you have any further requests.

Reviewer 2 Report

In this paper, the authors state that IoT end devices today have sufficient processing power and operating system software to pre-process measured data instead of distributing raw sensor data over the network. It is known in theory that the process scheduling has significant impact on the communication efficiency, in particular in operating systems that employ cooperative multitasking. The authors have run empirical tests to quantify these effects in different usage scenarios.

The paper is well written and easy to read. The scientific contribution is useful as the authors provide documented measurements and draw reasonable conclusions. The analysis of the graphs presented in Section 4 need a little more detail as they seem to have an artificial boundary that has not been explained so far.

Introduction

The introduction is somewhat difficult to read because first you state that IoT (end) devices take more and more processing tasks and then go back and state that processing is done in the cloud or on the edge, respectively. Although I agree with everything you say here, it is difficult to follow your line of  argument due to these seemingly contradicting points.

Moreover, citing your own paper [1] is a weak proof for your point. Including more references from other authors here would be useful.

Section 3

Although this is not topic of this paper it would be interesting for a reader to see how your proprietary MAC layer relates to modern, standardized scheduling modes such as TSCH?

Section 4

The experimental results in section 4 make me wonder if there are some boundaries in effect that are not explained by the RDC/PTDC. For example, in Figure 6, I would expect the average throughput 10% RDC and PS1 to be around (40 + frame overhead) * 10 * 8 bps >= 3200 bps?

You should also justify the choice of the 40 B payload size. Why this size, why is this parameter not modified as well?

Minor editorial issues

l 35 IP -> Internet Protocol (IP)

l 93 6LowPAN -> 6LoWPAN

l 98 like -> such as

(l98: "provide Internet connectivity" - not really. CoAP enables constrained devices to integrate with the public Internet on the application level but it provides by no means connectivity.)

l 186 in -> on

l 206 "..." -> "." (if you have more examples, name them, otherwise, the sentence ends here.)

l 272 analyzes -> analyses

l 304 nothing -> noting

l 351, 354 indent these bullet points as they are sub-items to l 349

References

[27] is missing a link to the actual data set; no match found in the mendeley database

Author Response

The authors would like to thank the reviewers for their time and their valuable comments. We really appreciate their effort and interest.

Following the reviewer 2’s suggestions and comments, we have made the following changes to our paper.

Point 1: The introduction is somewhat difficult to read because first you state that IoT (end) devices take more and more processing tasks and then go back and state that processing is done in the cloud or on the edge, respectively. Although I agree with everything you say here, it is difficult to follow your line of argument due to these seemingly contradicting points.

Moreover, citing your own paper [1] is a weak proof for your point. Including more references from other authors here would be useful.

Response 1: We really appreciate this suggestion. The first two sentences of the third paragraph of the introduction can cause some confusion. In them, we intend to refer to the IoT applications, not to the end-devices. We have rewritten them to make it clearer. This is the new version of the paragraph:

“On the other hand, IoT applications increasingly demand a more comprehensive understanding of the environment. That is, applications are no longer limited to obtaining direct measures of certain environmental variables through simple sensors, e.g., temperature, humidity or light sensors, but a higher level of information is sought through the processing of these raw data [1–3]. Performing this processing in the cloud is not desired because of latency issues and the generation of a massive network traffic due to the data uploading. For this reason, and because of the improvement of features and resources in the end-devices, these devices are increasingly being responsible for all or some of this processing. This trend of placing the computing and storage capabilities where the data is generated is referred to as edge and/or fog computing and it is gaining importance in the IoT [4,5]. However, delegating these storing and processing tasks to the end-devices increases again their computational load, and introduces other challenges in terms of resource and process management: real-time scheduling, storage allocation, energy consumption, among others.”

Point 2: Although this is not topic of this paper it would be interesting for a reader to see how your proprietary MAC layer relates to modern, standardized scheduling modes such as TSCH?

Response 2: In order to clarify that point, we have included a paragraph in Section 3 in which we explain that our custom-defined MAC may be comparable with other scheduled MACs, since all of them have strict timing-constraints:

“Although we use a custom-defined MAC to perform our tests, this MAC is similar to others typically used in wireless networks. Most of them follow scheduled schemes with strict timing constraints—such as the standard 802.15.4e that uses Time-Slotted Channel Hopping (TSCH) scheme—which makes the process management a critical issue. It is important to note that on these scheduled MAC schemes the communication performance may drastically degrade if these timing constraints are not met.”

Point 3: The experimental results in section 4 make me wonder if there are some boundaries in effect that are not explained by the RDC/PTDC. For example, in Figure 6, I would expect the average throughput 10% RDC and PS1 to be around (40 + frame overhead) * 10 * 8 bps >= 3200 bps?

Response 3: Due to this comment, we have realized that we lacked a definition of the time between packets used in our tests, since the reviewer has understood that the nodes send 10 packets per second.

In the paper we stated that when a packet is sent the test application waits 100 ms to start sending the next packet. However, a packet also needs a variable transmission time to be sent, which depends on the MAC schedule and the transceiver configuration. This way, the time between packets is the sum of this variable transmission time and the wait time of 100 ms. Therefore, the theoretical best-case throughput at application level —without errors and without counting header bytes— is:

Taking this into account, the boundaries that can be seen in the Figures 5 to 8 are coherent with this equation. Since the transmission time is variable for each packet, we can obtain its average value from the throughput values measured in our tests.

In order to define the time between packets, we have included the following paragraph in the methodology description of Section 3:

“Each test performed for each scenario lasts 3 minutes. In each one of them, node N1 is continuously sending packets from the top of the network stack, so all the layers described before are involved in the communication. In this way, each packet takes a variable time to be transmitted, which depends on the MAC schedule and the transceiver configuration. Then, when a packet has been sent, the node waits for 100 ms to start sending the next packet, and so on until the test time is over. Therefore, the time between packets is defined by the sum of the variable transmission time and this fixed wait time.”

Point 4: You should also justify the choice of the 40 B payload size. Why this size, why is this parameter not modified as well?

Response 4: We completely agree with this assessment. We have chosen 40 bytes of payload size as it is the maximum size allowed by the network stack. This stack was designed to be compatible with most radio transceivers and many of them have 64 bytes buffers. Therefore, 24 bytes are reserved for protocol headers while the rest may be used for payload.

In this work, we have carried out the tests using only this value —which optimizes the throughput— but it could be interesting to perform more tests varying this parameter. We have included a paragraph in the Conclusions section pointing to this possible future work. In addition, we have justified the 40 bytes payload election in the methodology description of Section 3 with this paragraph:

“Each packet sent is filled with 40 application bytes and contains a unique and incremental identifier. This is the maximum application payload length allowed by the network stack, since up to 24 bytes are reserved for protocol headers and the maximum buffer size of many radio transceivers is 64 bytes”

Point 5: Minor editorial issues.

Response 5: All the issues commented have been corrected in the manuscript file.

Point 6: References. [27] is missing a link to the actual data set; no match found in the mendeley database

Response 6: The dataset is not published yet. We are waiting for the manuscript final acceptation to publish the dataset together with it. The unpublished version may be accessed through this link:

https://data.mendeley.com/datasets/rxsdfg8ct9/draft?a=e0495adf-66cb-43c4-b78a-86acd527241a

Thank you for your collaboration and do not hesitate to contact us if you have any further requests.
